# Evolving Normalization-Activation Layers

**Hanxiao Liu[†], Andrew Brock[‡], Karen Simonyan[‡], Quoc V. Le[†]**
[†]Google Research, Brain Team   [‡]DeepMind
{hanxiaol,ajbrock,simonyan,qvl}@google.com

## Abstract

Normalization layers and activation functions are fundamental components in deep networks and typically co-locate with each other. Here we propose to design them using an automated approach. Instead of designing them separately, we unify them into a single tensor-to-tensor computation graph, and evolve its structure starting from basic mathematical functions. Examples of such mathematical functions are addition, multiplication and statistical moments. The use of low-level mathematical functions, in contrast to the use of high-level modules in mainstream NAS, leads to a highly sparse and large search space which can be challenging for search methods. To address the challenge, we develop efficient rejection protocols to quickly filter out candidate layers that do not work well. We also use multi-objective evolution to optimize each layer's performance across many architectures to prevent overfitting. Our method leads to the discovery of *EvoNorms*, a set of new normalization-activation layers with novel, and sometimes surprising structures that go beyond existing design patterns. For example, some EvoNorms do not assume that normalization and activation functions must be applied sequentially, nor need to center the feature maps, nor require explicit activation functions. Our experiments show that EvoNorms work well on image classification models including ResNets, MobileNets and EfficientNets but also transfer well to Mask R-CNN with FPN/SpineNet for instance segmentation and to BigGAN for image synthesis, outperforming BatchNorm and GroupNorm based layers in many cases.[1]

## 1  Introduction

Normalization layers and activation functions are fundamental building blocks in deep networks for stable optimization and improved generalization. Although they frequently co-locate, they are designed separately in previous works. There are several heuristics widely adopted during the design process of these building blocks. For example, a common heuristic for normalization layers is to use mean subtraction and variance division [1–4], while a common heuristic for activation functions is to use scalar-to-scalar transformations [5–11]. These heuristics may not be optimal as they treat normalization layers and activation functions as separate. Can automated machine learning discover a novel building block to replace these layers and go beyond the existing heuristics?

Here we revisit the design of normalization layers and activation functions using an automated approach. Instead of designing them separately, we unify them into a normalization-activation layer. With this unification, we can formulate the layer as a tensor-to-tensor computation graph consisting of basic mathematical functions such as addition, multiplication and cross-dimensional statistical moments. These low-level mathematical functions form a highly sparse and large search space, in contrast to mainstream NAS which uses high-level modules (e.g., Conv-BN-ReLU). To address the challenge of the size and sparsity of the search space, we develop novel rejection protocols to efficiently filter out candidate layers that do not work well. To promote strong generalization across different architectures, we use multi-objective evolution to explicitly optimize each layer's

| | |
|---|---|
| BN-ReLU | $\max\left(\frac{x-\mu_{b,w,h}(x)}{\sqrt{s^2_{b,w,h}(x)}}\gamma+\beta,0\right)$ |
| EvoNorm-B0 | $\frac{x}{\max\left(\sqrt{s^2_{b,w,h}(x)},v_1x+\sqrt{s^2_{w,h}(x)}\right)}\gamma+\beta$ |

Table 1: A searched layer named EvoNorm-B0 which consistently outperforms BN-ReLU. $\mu_{b,w,h}$, $s^2_{b,w,h}$, $s^2_{w,h}$, $v_1$ refer to batch mean, batch variance, instance variance and a learnable variable.

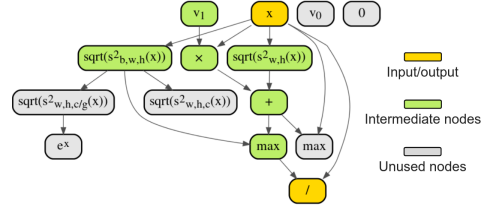

Figure 1: Computation graph of EvoNorm-B0.

performance over multiple architectures. Our method leads to the discovery of a set of novel layers, dubbed EvoNorms, with surprising structures that go beyond expert designs (an example layer is shown in Table 1 and Figure 1). For example, our most performant layers allow normalization and activation functions to interleave, in contrast to BatchNorm-ReLU or ReLU-BatchNorm [1,12] where normalization and activation function are applied sequentially. Some EvoNorms do not attempt to center the feature maps, and others require no explicit activation functions.

EvoNorms consist of two series: B series and S series. The B series are **b**atch-dependent and were discovered by our method without any constraint. The S series work on individual **s**amples, and were discovered by rejecting any batch-dependent operations. We verify their performance on a number of image classification architectures, including ResNets [13], MobileNetV2 [14] and EfficientNets [15]. We also study their interactions with a range of data augmentations, learning rate schedules and batch sizes, from 32 to 1024 on ResNets. Our experiments show that EvoNorms can substantially outperform popular layers such as BatchNorm-ReLU and GroupNorm-ReLU. On Mask R-CNN [16] with FPN [17] and with SpineNet [18], EvoNorms achieve consistent gains on COCO instance segmentation with negligible computation overhead. To further verify their generalization, we pair EvoNorms with a BigGAN model [19] and achieve promising results on image synthesis.

Our contributions can be summarized as follows:

- We are the first to search for the combination of normalization-activation layers. Our proposal to unify them into a single graph and the search space are novel. Our work tackles a missing link in AutoML by providing evidence that it is possible to use AutoML to discover a new building block from low-level mathematical operations (see Sec. 2 for a comparison with related works). A combination of our work with traditional NAS may realize the full potential of AutoML in automatically designing machine learning models from scratch.

- We propose novel rejection protocols to filter out candidates that do not work well or fail to work, based on both their performance and stability. We are also the first to address strong generalization of layers by pairing each candidate with multiple architectures and to explicitly optimize their cross-architecture performance. Our techniques can be used by other AutoML methods that have large, sparse search spaces and require strong generalization.

- Our discovered layers, EvoNorms, by themselves are novel contributions because they are different from previous works. They work well on a diverse set of architectures, including ResNets [12,13], MobileNetV2 [14], MnasNet [20], EfficientNets [15], and transfer well to Mask R-CNN [16], SpineNet [21] and BigGAN-deep [19]. E.g., on Mask-RCNN our gains are +1.9AP over BN-ReLU and +1.3AP over GN-ReLU. EvoNorms have a high potential impact because normalization and activation functions are central to deep learning.

- EvoNorms shed light on the design of normalization and activation layers. E.g., their structures suggest the potential benefits of non-centered normalization schemes, mixed variances and tensor-to-tensor rather than scalar-to-scalar activation functions (these properties can be seen in Table 1). Some of these insights can be used by experts to design better layers.

## 2 Related Works

Separate efforts have been devoted to design better activation functions and normalization layers, either manually [1–4, 7–10, 22, 23] or automatically [11, 24]. Different from the previous works, we eliminate the boundary between normalization and activation layers and search them jointly as a unified building block. Our search space is more challenging than those in the existing automated approaches [11, 24]. For example, we avoid relying on common heuristics like mean subtraction

or variance division in handcrafted normalization schemes; we search for general tensor-to-tensor transformations instead of scalar-to-scalar transformations in activation function search [11].

Our approach is inspired by recent works on neural architecture search, e.g., [20,25–50], but has a very different goal. While existing works aim to specialize an architecture built upon well-defined building blocks such as Conv-BN-ReLU or inverted bottlenecks [14], we aim to discover new building blocks starting from low-level mathematical functions. Our motivation is similar to AutoML-Zero [51] but has a more practical focus: the method not only leads to novel building blocks with new insights, but also achieves competitive results across many large-scale computer vision tasks.

Our work is also related to recent efforts on improving the initialization conditions for deep networks [52–54] in terms of challenging the necessity of traditional normalization layers. While those techniques are usually architecture-specific, our method discovers layers that generalize well across a variety of architectures and tasks without specialized initialization strategies.

## 3   Search Space

**Layer Representation.**   We represent each normalization-activation layer as a computation graph that transforms an input tensor into an output tensor (an example is shown in Figure 1). The computation graph is a DAG that has 4 initial nodes, including the input tensor and three auxiliary nodes: a constant zero tensor, and two trainable vectors $v_0$ and $v_1$ along the channel dimension initialized as 0's and 1's, respectively. In general, the DAG can have any number of nodes, but we restrict the total number of nodes to 4+10=14 in our experiments. Each intermediate node in the DAG represents the outcome of either a unary or a binary operation (shown in Table 2).

**Primitive Operations.**   Table 2 shows the primitive operations in the search space, including element-wise operations and aggregation operations that enable communication across different axes of the tensor.

Here we explain the notations $\mathcal{I}, \mu,$ and $s$ in the Table for aggregation ops. First, an aggregation op needs to know the axes (index set) where it can operate, which is denoted by $\mathcal{I}$. Let $x$ be a 4-dimensional tensor of feature maps. We use $b, w, h, c$ to refer to its batch, width, height and channel dimensions, respectively. We use $x_{\mathcal{I}}$ to represent a subset of $x$'s elements along the dimensions indicated by $\mathcal{I}$. For example, $\mathcal{I} = (b, w, h)$ indexes all the elements of $x$ along the batch, width and height dimensions; $\mathcal{I} = (w, h)$ refers to elements along the spatial dimensions only.

| Element-wise Ops | Expression | Arity |
|---:|:---|:---:|
| Add | $x + y$ | 2 |
| Mul | $x \times y$ | 2 |
| Div | $x/y$ | 2 |
| Max | $\max(x, y)$ | 2 |
| Neg | $-x$ | 1 |
| Sigmoid | $\sigma(x)$ | 1 |
| Tanh | $\tanh(x)$ | 1 |
| Exp | $e^x$ | 1 |
| Log | $\text{sign}(x) \cdot \ln(|x|)$ | 1 |
| Abs | $|x|$ | 1 |
| Square | $x^2$ | 1 |
| Sqrt | $\text{sign}(x) \cdot \sqrt{|x|}$ | 1 |

| Aggregation Ops | Expression | Arity |
|---:|:---|:---:|
| 1st order | $\mu_{\mathcal{I}}(x)$ | 1 |
| 2nd order | $\sqrt{\mu_{\mathcal{I}}(x^2)}$ | 1 |
| 2nd order, centered | $\sqrt{s_{\mathcal{I}}^2(x)}$ | 1 |

The notations $\mu$ and $s$ indicate ops that compute statistical moments, a natural way to aggregate over a set of elements. Let $\mu_{\mathcal{I}}(x)$ be a mapping that replaces each element in $x$ with the 1st order moment of $x_{\mathcal{I}}$. Likewise, let $s_{\mathcal{I}}^2(x)$ be a mapping that transforms each element of $x$ into the 2nd order moment among the elements in $x_{\mathcal{I}}$. Note both $\mu_{\mathcal{I}}$ and $s_{\mathcal{I}}^2$ preserve the shape of the original tensor.

Table 2: Search space primitives. The index set $\mathcal{I}$ can take any value among $\{(b, w, h), (w, h, c), (w, h), (w, h, c/g)\}$. A small $\epsilon$ is inserted as necessary for numerical stability. All the operations preserve the shape of the input tensor.

Finally, we use $\cdot/g$ to indicate that aggregation is carried out in a grouped manner along a dimension. We allow $\mathcal{I}$ to take values among $(b, w, h), (w, h, c), (w, h)$ and $(w, h, c/g)$. Combinations like $(b, w/g, h, c)$ or $(b, w, h/g, c)$ are not considered to ensure the model remains fully convolutional.

**Random Graph Generation.**   A random computation graph in our search space can be generated in a sequential manner. Starting from the initial nodes, we generate each new node by randomly sampling a primitive op and then randomly sampling its input nodes according to the op's arity. The process is repeated multiple times and the last node is used as the output.

With the above search space, we perform several small scale experiments with random sampling to understand its behaviors. Our observations are as follows:

**Observation 1: High sparsity of the search space.** While our search space can be expanded further, it is already large enough to be challenging. As an exercise, we took 5000 random samples from the search space and plugged them into three architectures on CIFAR-10. Figure 2 shows that none of the 5000 samples can outperform BN-ReLU. The accuracies for the vast majority of them are no better than random guess (note the y-axis is in log scale). A typical random layer would look like $\text{sign}(z)\sqrt{z}\gamma + \beta$, $z = \sqrt{s_{w,h}^2(\sigma(|x|))}$ and leads to near-zero ImageNet accuracies. Although random search does not seem to work well, we will demonstrate later that with a better search method, this search space is interesting enough to yield performant layers with highly novel structures.

**Observation 2: Weak generalization across architectures.** It is our goal to develop layers that work well across many architectures, e.g., ResNets, MobileNets, EfficientNets etc. We refer to this as *strong generalization* because it is a desired property of BatchNorm-ReLU. As another exercises, we pair each of the 5000 samples with three different architectures, and plot the accuracy calibrations on CIFAR-10. The results are shown in Figure 3, which indicate that layers that perform well on one architecture can fail completely on the other ones. Specifically, a layer that performs well on ResNets may not enable meaningful learning on MobileNets or EfficientNets at all.

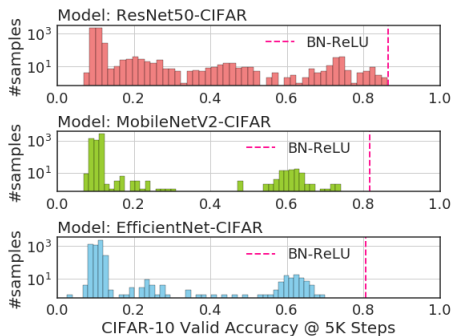

Figure 2: CIFAR-10 accuracy histograms of 5K random layers over three architectures.

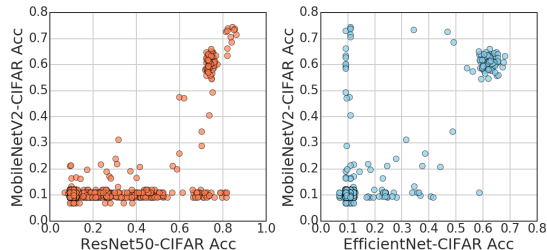

Figure 3: CIFAR-10 accuracy calibrations of 5K random layers over three architectures. The calibration is far from perfect. For example, layers performing well on ResNet-CIFAR may not be able to outperform random guess on MobileNetV2-CIFAR.

# 4  Search Method

In this section, we propose to use evolution as the search method (Sec. 4.1), and modify it to address the sparsity and achieve strong generalization. In particular, to address the sparsity of the search space, we propose efficient rejection protocols to filter out a large number of undesirable layers based on their quality and stability (Sec. 4.2). To achieve strong generalization, we propose to evaluate each layer over many different architectures and use multi-objective optimization to explicitly optimize its cross-architecture performance (Sec. 4.3).

## 4.1  Evolution

Here we propose to use evolution to search for better layers. The implementation is based on a variant of tournament selection [55]. At each step, a tournament is formed based on a random subset of the population. The winner of the tournament is allowed to produce offspring, which will be evaluated and added into the population. The overall quality of the population hence improves as the process repeats. We also regularize the evolution by maintaining a sliding window of only the most recent portion of the population [41]. To produce an offspring, we mutate the computation graph of the winning layer in three steps. First, we select an intermediate node uniformly at random. Then we replace its current operation with a new one in Table 2 uniformly at random. Finally, we select new predecessors for this node among existing nodes in the graph uniformly at random.

## 4.2  Rejection protocols to address high sparsity of the search space

Although evolution improves sample efficiency over random search, it does not resolve the high sparsity issue of the search space. This motivates us to develop two rejection protocols to filter out bad layers after short training. A layer must pass both tests to be considered by evolution.

**Quality.** We discard layers that achieve less than $20\%$[2] CIFAR-10 validation accuracy after training for 100 steps. Since the vast majority of the candidate layers attain poor accuracies, this simple mechanism ensures the compute resources to concentrate on the full training of a small number of promising candidates. Empirically, this speeds up the search by up to two orders of magnitude.

**Stability.** In addition to quality, we reject layers that are subject to numerical instability. The basic idea is to stress-test the candidate layer by *adversarially* adjusting the model weights $\theta$ towards the direction of maximizing the network's gradient norm. Formally, let $\ell(\theta, G)$ be the training loss of a model when paired with computation graph $G$ computed based on a small batch of images. Instability of training is reflected by the worst-case gradient norm: $\max_\theta \|\partial\ell(\theta, G)/\partial\theta\|_2$. We seek to maximize this value by performing gradient ascent along the direction of $\frac{\partial\|\partial\ell(\theta,G)/\partial\theta\|_2}{\partial\theta}$ up to 100 steps. Layers whose gradient norm exceeding $10^8$ are rejected. The stability test focuses on robustness because it considers the worst case, hence is complementary to the quality test. This test is highly efficient–gradients of many layers are forced to quickly blow up in less than 5 steps. We provide an ablation study on the effectiveness of the stability criterion in Appendix E.2.

### 4.3 Multi-architecture evaluation to promote strong generalization

To explicitly promote strong generalization, we formulate the search as a multi-objective optimization problem, where each candidate layer is always evaluated over multiple different *anchor* architectures to obtain a set of fitness scores. We choose three architectures as the anchors, including ResNet50 [13] (v2)[3], MobileNetV2 [14] and EfficientNet-B0 [15]. Widths and strides of these Imagenet architectures are adapted w.r.t the CIFAR-10 dataset [56] (Appendix B), on which they will be trained and evaluated for speedy feedback. The block definitions of the anchor architectures are shown in Figure 4.

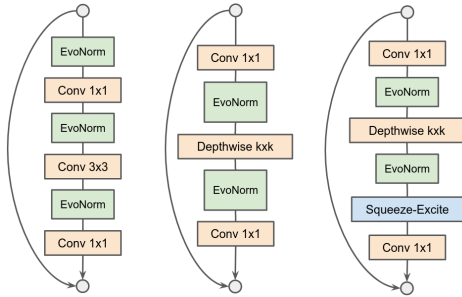

Figure 4: Block definitions of anchor architectures: ResNet-CIFAR (*left*), MobileNetV2-CIFAR (*center*), and EfficientNet-CIFAR (*right*).[4]

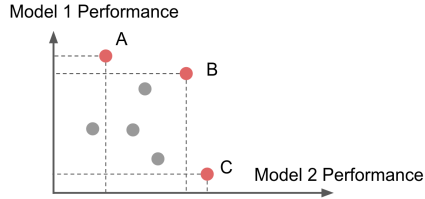

Figure 5: Illustration of tournament selection criteria for multi-objective optimization. Candidate B wins under the Average criterion. Each of A, B, C wins with probability $\frac{1}{3}$ under the Pareto criterion.

**Tournament Selection Criterion.** As each layer is paired with multiple architectures, and therefore has multiple scores, there are multiple ways to decide the tournament winner within evolution:

*Average*: Layer with the highest average accuracy wins (e.g., B in Figure 5 wins because it has the highest average performance on the two models.).

*Pareto*: A random layer on the Pareto frontier wins (e.g., A, B, C in Figure 5 are equal likely to win as none of them can be dominated by other candidates.).

Empirically we observe that different architectures have different accuracy variations. For example, ResNet has been observed to have a higher accuracy variance than MobileNet and EfficientNet on CIFAR-10 proxy tasks. Hence, under the Average criterion, the search will bias towards helping ResNet, not MobileNet nor EfficientNet. We therefore propose to use the Pareto frontier criterion to avoid this bias. Our method is novel, but reminiscent of NSGA-II [57], a well-established multi-objective genetic algorithm, in terms of simultaneously optimizing all the non-dominated solutions.

# 5 Experiments

We include experimental details in Appendix C, including those for the proxy task, search, reranking, and full-fledged evaluations. In summary, we did the search on CIFAR-10, and re-ranked the top-10 layers on a held-out set of ImageNet to obtain the best 3 layers. The top-10 layers are reported in Appendix D. Evolution on CIFAR-10 took 2 days to complete with 5000 CPU workers. For all results below, our layers and the baselines are compared under identical training setup. Hyperparameters are inherited from the original implementations (usually in favor of BNs) without tuning w.r.t EvoNorms.

## 5.1 Generalization across Image Classifiers

**Batch-dependent Layers.** In Table 3, we compare the three discovered layers against some widely used normalization-activation layers on ImageNet, including strong baselines with the SiLU/Swish activation function [9, 11, 22]. We refer to our layers as the EvoNorm-B series, as they involve **B**atch aggregations ($\mu_{b,w,h}$ and $s^2_{b,w,h}$) hence require maintaining a moving average statistics for inference. The table shows that EvoNorms are no worse than BN-ReLU across all cases, and perform better on average than the strongest baseline. It is worth emphasizing that hyperparameters and architectures used in the table are implicitly optimized for BatchNorms due to historical reasons.

| Layer | Expression | R-50 | | | | MV2 | MN | EN-B0 | EN-B5 |
|---|---|---|---|---|---|---|---|---|---|
| | | original | +aug | +aug+2× | +aug+2×+cos | | | | |
| BN-ReLU | $\max(z,0), \frac{x-\mu_{b,w,h}(x)}{\sqrt{s^2_{b,w,h}(x)}}\gamma+\beta$ | $76.3_{\pm0.1}$ | $76.2_{\pm0.1}$ | $77.6_{\pm0.1}$ | $77.7_{\pm0.1}$ | $73.4_{\pm0.1}$ | $74.6_{\pm0.2}$ | $76.4$ | $\mathbf{83.6}$ |
| BN-SiLU/Swish | $z\sigma(v_1 z), z=\frac{x-\mu_{b,w,h}(x)}{\sqrt{s^2_{b,w,h}(x)}}\gamma+\beta$ | $\mathbf{76.6}_{\pm0.1}$ | $77.3_{\pm0.1}$ | $\mathbf{78.2}_{\pm0.1}$ | $78.2_{\pm0.0}$ | $74.5_{\pm0.1}$ | $\mathbf{75.3}_{\pm0.1}$ | $\mathbf{77.0}$ | $83.5$ |
| Random | $\text{sign}(z)\sqrt{z}\gamma+\beta, z=\sqrt{s^2_{w,h}(\sigma(|x|))}$ | 1e-3 | 1e-3 | 1e-3 | 1e-3 | 1e-3 | 1e-3 | 1e-3 | 1e-3 |
| Random + rej | $\tanh(\max(x,\tanh(x)))\gamma+\beta$ | $71.7_{\pm0.2}$ | $70.8_{\pm0.1}$ | $63.6_{\pm18.9}$ | $55.3_{\pm17.5}$ | 1e-3 | 1e-3 | 1e-3 | 1e-3 |
| RS + rej | $\frac{\max(x,0)}{\sqrt{\mu_{b,h}(x^2)}}\gamma+\beta$ | $75.8_{\pm0.1}$ | $76.3_{\pm0.0}$ | $77.4_{\pm0.1}$ | $77.5_{\pm0.1}$ | $73.5_{\pm0.1}$ | $74.6_{\pm0.1}$ | $76.4$ | $83.2$ |
| EvoNorm-B0 | $\frac{x}{\max\left(\sqrt{s^2_{b,w,h}(x)},v_1 x+\sqrt{s^2_{w,h}(x)}\right)}\gamma+\beta$ | $\mathbf{76.6}_{\pm0.0}$ | $77.7_{\pm0.1}$ | $77.9_{\pm0.1}$ | $\mathbf{78.4}_{\pm0.1}$ | $\mathbf{75.0}_{\pm0.1}$ | $\mathbf{75.3}_{\pm0.0}$ | $76.8$ | $\mathbf{83.6}$ |
| EvoNorm-B1 | $\frac{x}{\max\left(\sqrt{s^2_{b,w,h}(x)},(x+1)\sqrt{\mu_{w,h}(x^2)}\right)}\gamma+\beta$ | $76.1_{\pm0.1}$ | $77.5_{\pm0.0}$ | $77.7_{\pm0.0}$ | $78.0_{\pm0.1}$ | $74.6_{\pm0.1}$ | $75.1_{\pm0.1}$ | $76.5$ | $\mathbf{83.6}$ |
| EvoNorm-B2 | $\frac{x}{\max\left(\sqrt{s^2_{b,w,h}(x)},\sqrt{\mu_{w,h}(x^2)}-x\right)}\gamma+\beta$ | $\mathbf{76.6}_{\pm0.2}$ | $77.7_{\pm0.1}$ | $78.0_{\pm0.1}$ | $\mathbf{78.4}_{\pm0.1}$ | $74.6_{\pm0.1}$ | $75.0_{\pm0.1}$ | $76.6$ | $83.4$ |

Table 3: ImageNet results of batch-dependent normalization-activation layers. Terms requiring moving average statistics are highlighted in blue. Each layer is evaluated on ResNets (R), MobileNetV2 (MV2), MnasNet-B1 (MN) and EfficientNets (EN). We also vary the training settings for ResNet-50: "aug", "2×" and "cos" refer to the use of RandAugment [58], longer training (180 epochs instead of 90) and cosine learning rate schedule [59], respectively. Results in the same column are obtained using identical training setup. Entries with error bars are aggregated over three independent runs.

Table 3 also shows that a random layer in our search space only achieves near-zero accuracy on ImageNet. It then shows that with our proposed rejection rules in Sec. 4.2 (Random + rej), one can find a layer with meaningful accuracies on ResNets. Finally, using comparable compute with evolution, random search with rejection (RS + rej) can discover a compact variant of BN-ReLU. This layer achieves promising results across all architectures, albeit clearly worse than EvoNorms. The search progress of random search relative to evolution is shown in Figure 6 and in Appendix E.

**Batch-independent Layers.** Table 4 presents EvoNorms obtained from another search experiment, during which layers containing batch aggregation ops are excluded. The goal is to design layers that rely on individual samples only, a desirable property to simplify implementation and to stabilize training with small batch sizes. We refer to these **S**ample-based layers as the EvoNorm-S series. We compare them against handcrafted baselines designed under a similar motivation, including Group Normalization [4] (GN-ReLU) and a recently proposed layer aiming to eliminate batch dependencies [23] (FRN). Table 4 and its accompanying figure show that EvoNorm-S layers achieve competitive or better results than all the baselines across a wide range of batch sizes.

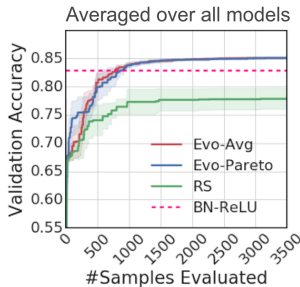

Figure 6: Search progress on CIFAR-10. Each curve is averaged over the top-10 layers in the population and over the three anchor architectures. Only samples survived the quality and stability tests are considered.

| Layer | Expression | ResNet-50 | | | | MobileNetV2 | |
|---|---|---|---|---|---|---|---|
| | | original | | +aug+2×+cos | | | |
| | | large 128×8 | small 4×8 | large 128×8 | small 4×8 | large 128×32 | small 4×32 |
| BN-ReLU | $\max(z,0), z = \frac{x-\mu_{b,w,h}(x)}{\sqrt{s^2_{b,w,h}(x)}}\gamma + \beta$ | $\mathbf{76.3}_{\pm0.1}$ | $70.9_{\pm0.3}$ | $77.7_{\pm0.1}$ | $70.5_{\pm1.1}$ | $73.4_{\pm0.1}$ | $66.7_{\pm1.9}$ |
| GN-ReLU | $\max(z,0), z = \frac{x-\mu_{w,h,c/g}(x)}{\sqrt{s^2_{w,h,c/g}(x)}}\gamma + \beta$ | $75.3_{\pm0.1}$ | $75.8_{\pm0.1}$ | $77.1_{\pm0.1}$ | $77.2_{\pm0.1}$ | $72.2_{\pm0.1}$ | $72.4_{\pm0.1}$ |
| FRN | $\max(z,v_0), z = \frac{x}{\sqrt{\mu_{w,h}(x^2)}}\gamma + \beta$ | $75.6_{\pm0.1}$ | $75.9_{\pm0.1}$ | $54.7_{\pm14.3}$ | $77.4_{\pm0.1}$ | $73.4_{\pm0.2}$ | $73.5_{\pm0.1}$ |
| EvoNorm-S0 | $\frac{x\sigma(v_1 x)}{\sqrt{s^2_{w,h,c/g}(x)}}\gamma + \beta$ | $76.1_{\pm0.1}$ | $\mathbf{76.5}_{\pm0.1}$ | $\mathbf{78.3}_{\pm0.1}$ | $\mathbf{78.3}_{\pm0.1}$ | $\mathbf{73.9}_{\pm0.2}$ | $\mathbf{74.0}_{\pm0.1}$ |
| EvoNorm-S1 | $\frac{x\sigma(x)}{\sqrt{s^2_{w,h,c/g}(x)}}\gamma + \beta$ | $76.1_{\pm0.1}$ | $76.3_{\pm0.1}$ | $78.2_{\pm0.1}$ | $78.2_{\pm0.1}$ | $73.6_{\pm0.1}$ | $73.7_{\pm0.1}$ |
| EvoNorm-S2 | $\frac{x\sigma(x)}{\sqrt{\mu_{w,h,c/g}(x^2)}}\gamma + \beta$ | $76.0_{\pm0.1}$ | $76.3_{\pm0.1}$ | $77.9_{\pm0.1}$ | $78.0_{\pm0.1}$ | $73.7_{\pm0.1}$ | $73.8_{\pm0.1}$ |

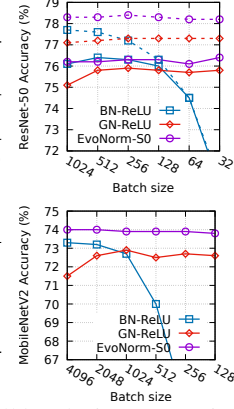

Table 4: *Left*: ImageNet results of batch-independent layers with large and small batch sizes. Learning rates are scaled linearly relative to the batch sizes [60]. For ResNet-50 we report results under both the standard training setting and the fancy setting (+aug+2×+cos). We also report full results under four different training settings in Appendix E. *Right*: Performance as the batch size decreases. For ResNet, solid and dashed lines are obtained with standard setting and fancy setting, respectively.

## 5.2 Generalization to Instance Segmentation

To investigate if our discovered layers generalize beyond the classification task that we searched on, we pair them with Mask R-CNN [16] for object detection and instance segmentation on COCO [61]. We consider two different types of backbones: ResNet-FPN [17] and SpineNet [21]. The latter is particularly interesting because the architecture has a highly non-linear layout which is very different from the classification models used during search. In all of our experiments, EvoNorms are applied to both the backbone and the heads to replace their original activation-normalization layers. Detailed training settings of these experiments are available in Appendix C.

| Backbone | Layer | $AP^{bbox}$ | $AP^{bbox}_{50}$ | $AP^{bbox}_{75}$ | $AP^{mask}$ | $AP^{mask}_{50}$ | $AP^{mask}_{75}$ | MAdds (B) | Params (M) | Batch Indep. |
|---|---|---|---|---|---|---|---|---|---|---|
| R-50-FPN | BN-ReLU | 42.1 | 62.9 | 46.2 | 37.8 | 60.0 | 40.6 | 379.7 | 46.37 | ✗ |
| | BN-SiLU/Swish | (+1.0)43.1 | 63.8 | 47.3 | (+0.6)38.4 | 60.6 | 41.4 | 379.9 | 46.38 | |
| | EvoNorm-B0 | (+1.9)**44.0** | **65.2** | **48.1** | (+1.7)**39.5** | **62.7** | **42.4** | 380.4 | 46.38 | |
| | GN-ReLU | 42.7 | 63.8 | 46.6 | 38.4 | 61.2 | 41.2 | 380.8 | 46.37 | ✓ |
| | EvoNorm-S0 | (+0.9)**43.6** | **64.9** | **47.9** | (+1.0)**39.4** | **62.3** | **42.7** | 380.4 | 46.38 | |
| SpineNet-96 | BN-ReLU | 47.1 | 68.0 | 51.5 | 41.5 | 65.0 | 44.3 | 315.0 | 55.19 | ✗ |
| | BN-SiLU/Swish | (+0.5)47.6 | 68.2 | 52.0 | (+0.5)42.0 | 65.6 | 45.5 | 315.2 | 55.21 | |
| | EvoNorm-B0 | (+0.9)**48.0** | **68.7** | **52.6** | (+0.9)**42.4** | **66.2** | **45.8** | 315.8 | 55.21 | |
| | GN-ReLU | 45.7 | 66.8 | 49.9 | 41.0 | 64.3 | 43.9 | 316.5 | 55.19 | ✓ |
| | EvoNorm-S0 | (+1.8)**47.5** | **68.5** | **51.8** | (+1.1)**42.1** | **65.9** | **45.3** | 315.8 | 55.21 | |

Table 5: Mask R-CNN object detection and instance segmentation results on COCO `val2017`.

Results are summarized in Table 5. With both ResNet-FPN and SpineNet backbones, EvoNorms significantly improve the APs with negligible impact on FLOPs or model sizes. While EvoNorm-B0 offers the strongest results, EvoNorm-S0 outperforms commonly used layers, including GN-ReLU and BN-ReLU, by a clear margin without requiring moving-average statistics. These results demonstrate strong generalization beyond the classification task that the layers were searched on.

## 5.3 Generalization to GAN Training

We further test the applicability of EvoNorms to training GANs [62]. Normalization is particularly important because the unstable dynamics of the adversarial game render training sensitive to nearly every aspect of its setup. We replace the BN-ReLU layers in the generator of BigGAN-deep [19] with EvoNorms and with previously designed layers, and measure performance on ImageNet generation at 128×128

| Layer | IS (median/best) | FID (median/best) |
|---|---|---|
| BN-ReLU | **118.77/124.01** | 7.85/7.29 |
| EvoNorm-B0 | 101.13/113.63 | **6.91/5.87** |
| GN-ReLU | 99.09 | 8.14 |
| LayerNorm-ReLU | 91.56 | 8.35 |
| PixelNorm-ReLU | 88.58 | 10.41 |
| EvoNorm-S0 | **104.64/113.96** | **6.86/6.26** |

Table 6: BigGAN-deep results w. batch-dependent and batch-independent layers. We report median and best performance across 3 random seeds.

resolution using Inception Score (IS, [63]) and Fréchet Inception distance (FID, [64]). We compare two of our most performant layers, B0 and S0, against the baseline BN-ReLU and GN-ReLU, as well as LayerNorm-ReLU [2], and PixelNorm-ReLU [65], a layer designed for a different GAN architecture. We sweep the number of groups in GN-ReLU from 8,16,32, and report results using 16 groups. Consistent with BigGAN training, we report results at peak performance in Table 6. Note higher is better for IS, lower is better for FID.

Swapping BN-ReLU out for most other layers substantially cripples training, but both EvoNorm-B0 and S0 achieve comparable, albeit worse IS, and improved FIDs over the BN-ReLU baseline. Notably, EvoNorm-S0 outperforms all the other per-sample normalization-activation layers in both IS and FID. This result further confirms that EvoNorms transfer to visual tasks in multiple domains.

## 5.4  Intriguing Properties of EvoNorms

**EvoNorm-B0.**  Unlike conventional normalization schemes relying on a single type of variance only, EvoNorm-B0 mixes together two types of statistical moments in its denominator, namely $s_{b,w,h}^2(x)$ (batch variance) and $s_{w,h}^2(x)$ (instance variance). The former captures global information across images in the same mini-batch, and the latter captures local information per image. It is also interesting to see that B0 does not have any explicit activation function because of its intrinsically nonlinear normalization process. Table 7 shows that manual modifications to the structure of B0 can substantially cripple the training, demonstrating its local optimality.

| Expression | Modification | Accuracy (%) | | |
|---|---|---|---|---|
| $\frac{x}{\max\left(\sqrt{s_{b,w,h}^2(x)}, v_1 x + \sqrt{s_{w,h}^2(x)}\right)}$ | None | 76.5 | 76.6 | 76.6 |
| $\frac{x}{\max\left(\sqrt{s_{b,w,h}^2(x)}, \sqrt{s_{w,h}^2(x)}\right)}$ | No $v_1 x$ | 14.4 | 50.5 | 22.0 |
| $\frac{x}{\sqrt{s_{b,w,h}^2(x)}}$ | No local term | 4.2 | 4.1 | 4.1 |
| $\frac{x}{v_1 x + \sqrt{s_{w,h}^2(x)}}$ | No global term | 1e-3 | 1e-3 | 1e-3 |
| $\frac{x}{\sqrt{s_{b,w,h}^2(x)} + v_1 x + \sqrt{s_{w,h}^2(x)}}$ | max → add | 1e-3 | 1e-3 | 1e-3 |

Table 7: Impact of structural changes to EvoNorm-B0. For each variant we report its ResNet-50 ImageNet accuracies over three random seeds at the point when NaN error (if any) occurs.

| Layer | R-50 | MV2 |
|---|---|---|
| BN-ReLU | $70.9_{\pm 0.3}$ | $66.7_{\pm 1.9}$ |
| GN-ReLU | $75.8_{\pm 0.1}$ | $72.4_{\pm 0.1}$ |
| GN-SiLU/Swish | $\mathbf{76.5}_{\pm 0.0}$ | $73.1_{\pm 0.1}$ |
| EvoNorm-S0 | $\mathbf{76.5}_{\pm 0.1}$ | $\mathbf{74.0}_{\pm 0.1}$ |

Table 8: ImageNet classification accuracies of ResNet-50 and MobileNetV2 with 4 images/worker.

**EvoNorm-S0.**  It is interesting to observe the SiLU/Swish activation function [9, 11, 22] as a part of EvoNorm-S0. The algorithm also learns to divide the post-activation features by the standard deviation part of GroupNorm [4] (GN). Note this is not equivalent to applying GN and SiLU/Swish sequentially. The full expression for GN-SiLU/Swish is $\frac{x - \mu_{w,h,c/g}}{\sqrt{s_{w,h,c/g}^2(x)}} \sigma\left(v_1 \frac{x - \mu_{w,h,c/g}}{\sqrt{s_{w,h,c/g}^2(x)}}\right)$ whereas the expression for S0 is $\frac{x}{\sqrt{s_{w,h,c/g}^2(x)}} \sigma(v_1 x)$ (omitting $\gamma$ and $\beta$). The latter is more compact and efficient.

The overall structure of EvoNorm-S0 offers an interesting hint that SiLU/Swish-like nonlinearities and grouped normalizers may be complementary with each other. Although both GN and Swish have been popular in the literature, their combination is under-explored to the best of our knowledge. In Table 8 we evaluate GN-SiLU/Swish and compare it with other layers that are batch-independent. The results confirm that both EvoNorm-S0 and GN-SiLU/Swish can indeed outperform GN-ReLU by a clear margin, though EvoNorm-S0 generalizes better on MobileNetV2.

**Scale Invariance.**  Interestingly, most EvoNorms attempt to promote scale-invariance, an intriguing property from the optimization perspective [66, 67]. See Appendix E.3 for more analysis.

## 6  Conclusion

In this work, we unify normalization layer and activation function as single tensor-to-tensor computation graph consisting of basic mathematical functions. Unlike mainstream NAS works that specialize a network based on existing layers (Conv-BN-ReLU), we aim to discover new layers that can generalize well across many different architectures. We first identify challenges including high search space sparsity and the weak generalization issue. We then propose techniques to overcome these challenges using efficient rejection protocols and multi-objective evolution. Our method discovered novel layers with surprising structures that achieve strong generalization across many architectures and tasks.

## Broader Impact

Since normalization-activation layers are critical components in state-of-the-art neural networks, we expect that the discovered modules to benefit a wide range of deep learning applications and yield positive impacts on healthcare, autonomous driving, manufacturing, agriculture and more. Insights derived from these layers may also deepen the community's understanding about the optimization properties of neural networks hence result in theoretical advancements. The proposed layer search method can be used as a tool to discover new fundamental building blocks besides normalization-activation layers, accelerating scientific discovery about novel machine learning concepts in general. On the negative side, the layer search process requires a relatively large number of CPU cores hence may lead to increased carbon footprint over the manual approaches.

## Acknowledgements and Disclosure of Funding

The authors would like to thank Gabriel Bender, Chen Liang, Esteban Real, Sergey Ioffe, Prajit Ramachandran, Pengchong Jin, Xianzhi Du, Ekin D. Cubuk, Barret Zoph, Da Huang, and Mingxing Tan for their comments and support.

This work was done as a part of the authors' full-time jobs at Google and DeepMind.

## Footnotes

[1]Code for EvoNorms on ResNets: https://github.com/tensorflow/tpu/tree/master/models/official/resnet

[2]This is twice as good as random guess on CIFAR-10.

[3]We always use the v2 instantiation of ResNets [13] where ReLUs are adjacent to BatchNorm layers.

[4]For each model, a custom layer is used to replace BN-ReLU/SiLU/Swish in the original architecture. Each custom layer is followed by a channel-wise affine transform. See pseudocode in Appendix A for details.

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
