[Supplementary Material]

# A   Code Snippets in TensorFlow

The following pseudocode relies on broadcasting to make sure the tensor shapes are compatible.

### BN-ReLU

```
def batchnorm_relu(x, gamma, beta, nonlinearity, training):
  mean, std = batch_mean_and_std(x, training)
  z = (x − mean) / std * gamma + beta
  if nonlinearity:
    return tf.nn.relu(z)
  else:
    return z
```

### EvoNorm-B0 (use this to replace BN-ReLU)

```
def evonorm_b0(x, gamma, beta, nonlinearity, training):
  if nonlinearity:
    v = trainable_variable_ones(shape=gamma.shape)
    _, batch_std = batch_mean_and_std(x, training)
    den = tf.maximum(batch_std, v * x + instance_std(x))
    return x / den * gamma + beta
  else:
    return x * gamma + beta
```

### EvoNorm-S0 (use this to replace BN-ReLU)

```
def evonorm_s0(x, gamma, beta, nonlinearity):
  if nonlinearity:
    v = trainable_variable_ones(shape=gamma.shape)
    num = x * tf.nn.sigmoid(v * x)
    return num / group_std(x) * gamma + beta
  else:
    return x * gamma + beta
```

### Helper functions for EvoNorms

```
def instance_std(x, eps=1e−5):
  _, var = tf.nn.moments(x, axes=[1, 2], keepdims=True)
  return tf.sqrt(var + eps)

def group_std(x, groups=32, eps=1e−5):
  N, H, W, C = x.shape
  x = tf.reshape(x, [N, H, W, groups, C // groups])
  _, var = tf.nn.moments(x, [1, 2, 4], keepdims=True)
  std = tf.sqrt(var + eps)
  std = tf.broadcast_to(std, x.shape)
  return tf.reshape(std, [N, H, W, C])

def trainable_variable_ones(shape, name="v"):
  return tf.get_variable(name, shape=shape,
            initializer=tf.ones_initializer())
```

# B   Proxy Task and Anchor Architectures

An ideal proxy task should be lightweight enough to allow speedy feedback. At the same time, the anchor architectures must be sufficiently challenging to train from the optimization point of view to stress-test the layers. These motivated us to consider a small training task and deep architectures.

We therefore define the proxy task as image classification on the CIFAR-10 dataset [56], and consider three representative ImageNet architectures (that are deep enough) adapted with respect to this setup, including Pre-activation ResNet50 [13] with channel multiplier $0.25\times$, MobileNetV2 [14] with channel multiplier $0.5\times$ and EfficientNet-B0 [15] with channel multiplier $0.5\times$. To handle the reduced image resolution on CIFAR-10 relative to ImageNet, the first two convolutions with spatial reduction are modified to use stride one for all of these architectures. We refer to these adapted versions as ResNet50-CIFAR, MobileNetV2-CIFAR and EfficientNet-CIFAR, respectively. We do not change anything beyond channel multipliers and strides described above, to preserve the original attributes (e.g., depth) of these ImageNet models as much as possible.

## C   Implementation Details

**Proxy Task on CIFAR-10.**   We use the same training setup for all the architectures. Specifically, we use 24×24 random crops on CIFAR-10 with a batch size of 128 for training, and use the original 32×32 image with a batch size of 512 for validation. We use SGD with learning rate 0.1, Nesterov momentum 0.9 and weight decay $10^{-4}$. Each model is trained for 2000 steps with a constant learning rate for the EvoNorm-B experiment. Each model is trained for 5000 steps following a cosine learning rate schedule for the EvoNorm-S experiment. These are chosen to ensure the majority of the models can achieve reasonable convergence quickly. With our implementation, it takes 3-10 hours to train each model on a single CPU worker with two cores.

**Evolution.**   We regularize the evolution [41] by considering a sliding window of only the most recent 2500 genotypes. Each tournament is formed by a random subset of 5% of the active population. Winner is determined as a random candidate on the Pareto-frontier w.r.t. the three accuracy scores (Sec. 4.1), which will be mutated twice in order to promote structural diversity. To encourage exploration, we further inject noise into the evolution process by replacing the offspring with a completely new random architecture with probability 0.5. Each search experiment takes 2 days to complete with 5000 CPU workers.

**Reranking.**   After search, we take the top-10 candidates from evolution and pair each of them with fully-fledged ResNet-50, MobileNetV2 and EfficientNet-B0. We then rerank the layers based on their averaged ImageNet accuracies of the three models. To avoid overfitting the validation/test metric, each model is trained using 90% of the ImageNet *training* set and evaluated over the rest of the 10%. The top-3 reranked layers are then used to obtain our main results. The reranking task is more computationally expensive than the proxy task we search with, but is accordingly more representative of the downstream tasks of interest, allowing for better distinguishing between top candidates.

**Classification on ImageNet.**   For ImageNet results presented in Table 3 (layers with batch statistics), we use a base learning rate of 0.1 per 256 images for ResNets, MobileNetV2 and MnasNet, and a base learning rate of 0.016 per 256 images for EfficientNets following the official implementation. Note these learning rates have been heavily optimized w.r.t. batch normalization. For results presented in Table 4 (layers without batch statistics), the base learning rate for MobileNetV2 is lowered to 0.03 (tuned w.r.t. GN-ReLU among 0.01, 0.03, 0.1). We use the standard multi-stage learning rate schedule in the basic training setting for ResNets, cosine schedule [59] for MobileNetV2 and MnasNet, and the original polynomial schedule for EfficientNets. These learning rate schedules also come with a linear warmup phase [60]. For all architectures, the batch size per worker is 128 and the input resolution is 224×224. The only exception is EfficientNet-B5, which uses batch size 64 per worker and input resolution 456×456. The number of workers is 8 for ResNets and 32 for the others. We use 32 groups for grouped aggregation ops $\mu_{w,h,c/g}$ and $s^2_{w,h,c/g}$.

**Instance Segmentation on COCO**   The training is carried out over 8 workers with 8 images/worker and the image resolution is 1024×1024. The models are trained from scratch for 135K steps using SGD with momentum 0.9, weight decay 4e-5 and an initial learning rate of 0.1, which is reduced by 10× at step 120K and step 125K. For SpineNet experiments, we follow the training setup of the original paper [21].

## D   Candidate layers without Reranking

### D.1   Top-10 EvoNorm-B candidates

1. $\dfrac{x}{\max\left(\sqrt{s^2_{b,w,h}(x)},z\right)}\gamma + \beta$, $z = (x+1)\sqrt{\mu_{w,h}(x^2)}$

2. $\dfrac{x}{\max\left(\sqrt{s^2_{b,w,h}(x)},z\right)}\gamma + \beta$, $z = x + \sqrt{\mu_{w,h,c}(x^2)}$

3. $-\dfrac{x\sigma(x)}{\sqrt{s^2_{b,w,h}(x)}}\gamma + \beta$

4. $\dfrac{x}{\max\left(\sqrt{s^2_{b,w,h}(x)},z\right)}\gamma + \beta$, $z = x\sqrt{\mu_{w,h}(x^2)}$

5. $\dfrac{x}{\max\left(\sqrt{s^2_{b,w,h}(x)},z\right)}\gamma + \beta,\ z = \sqrt{\mu_{w,h,c}(x^2)} - x$

6. $\dfrac{x}{\max\left(\sqrt{s^2_{b,w,h}(x)},z\right)}\gamma + \beta,\ v_1 x + \sqrt{s^2_{w,h}(x)}$

7. $\dfrac{x}{\max\left(\sqrt{s^2_{b,w,h}(x)},z\right)}\gamma + \beta,\ z = \sqrt{\mu_{w,h}(x^2)} - x$

8. $\dfrac{x}{\max\left(\sqrt{s^2_{b,w,h}(x)},z\right)}\gamma + \beta,\ z = x\sqrt{\mu_{w,h}(x^2)}$ (duplicate)

9. $\dfrac{x}{\max\left(\sqrt{s^2_{b,w,h}(x)},z\right)}\gamma + \beta,\ z = x + \sqrt{s^2_{w,h}(x)}$

10. $\dfrac{x}{\max\left(\sqrt{s^2_{b,w,h}(x)},z\right)}\gamma + \beta,\ z = x + \sqrt{s^2_{w,h}(x)}$ (duplicate)

## D.2   Top-10 EvoNorm-S candidates

1. $\dfrac{x\tanh(\sigma(x))}{\sqrt{\mu_{w,h,c/g}(x^2)}}\gamma + \beta$

2. $\dfrac{x\sigma(x)}{\sqrt{\mu_{w,h,c/g}(x^2)}}\gamma + \beta$

3. $\dfrac{x\sigma(x)}{\sqrt{\mu_{w,h,c/g}(x^2)}}\gamma + \beta$ (duplicate)

4. $\dfrac{x\sigma(x)}{\sqrt{\mu_{w,h,c/g}(x^2)}}\gamma + \beta$ (duplicate)

5. $\dfrac{x\sigma(x)}{\sqrt{s^2_{w,h,c/g}(x)}}\gamma + \beta$

6. $\dfrac{x\sigma(v_1 x)}{\sqrt{s^2_{w,h,c/g}(x)}}\gamma + \beta$

7. $\dfrac{x\sigma(x)}{\sqrt{s^2_{w,h,c/g}(x)}}\gamma + \beta$ (duplicate)

8. $\dfrac{x\sigma(x)}{\sqrt{\mu_{w,h,c/g}(x^2)}}\gamma + \beta$ (duplicate)

9. $\dfrac{x\sigma(x)}{\sqrt{s^2_{w,h,c/g}(x)}}\gamma + \beta$ (duplicate)

10. $z\sigma(\max(x,z))\gamma + \beta,\ z = \dfrac{x}{\sqrt{\mu_{w,h,c/g}(x^2)}}$

# E   Additional Results

Figure 7: Search progress of evolution vs. random search vs. a fixed baseline (BN-ReLU) on the proxy task. Each curve denotes the mean and standard deviation of the top-10 architectures in the population. Only valid samples survived the rejection phase are reported.

Figure 8: Training/eval curves for ResNet-50 (+aug) and MobileNetV2 on ImageNet with large batch sizes. The corresponding test accuracy for each layer is reported in the legend.

| Layer | Images / Worker | | | | | |
|---|---|---|---|---|---|---|
| | 128 | 64 | 32 | 16 | 8 | 4 |
| BN-ReLU | 76.1 | **76.4** | 76.3 | 76.0 | 74.5 | 70.4 |
| GN-ReLU | 75.1 | 75.8 | 75.9 | 75.8 | 75.7 | 75.8 |
| FRN | 75.5 | 75.8 | 75.9 | 75.8 | 75.8 | 75.8 |
| EvoNorm-S0 | 76.2 | 76.2 | 76.3 | **76.3** | 76.1 | **76.4** |
| EvoNorm-S1 | **76.3** | 76.1 | 76.2 | 76.2 | 76.1 | **76.4** |
| EvoNorm-S2 | 76.1 | 76.3 | **76.4** | 76.3 | 76.3 | **76.4** |

Table 9: ResNet-50 results as the batch size decreases (90 training epochs, original learning rate schedule).

| Layer | Images / Worker | | | | | |
|---|---|---|---|---|---|---|
| | 128 | 64 | 32 | 16 | 8 | 4 |
| BN-ReLU | 76.1 | 76.2 | 76.1 | 75.6 | 74.3 | 70.7 |
| GN-ReLU | 75.4 | 75.7 | 75.9 | 75.8 | 75.8 | 75.7 |
| FRN | 75.3 | 75.8 | 75.9 | 75.9 | 75.8 | 76.0 |
| EvoNorm-S0 | 77.4 | **77.4** | **77.6** | 77.3 | **77.3** | **77.5** |
| EvoNorm-S1 | **77.5** | 77.3 | 77.4 | **77.4** | **77.3** | 77.4 |
| EvoNorm-S2 | 76.9 | 77.3 | 77.1 | 77.2 | 77.1 | 77.1 |

Table 10: ResNet-50 results as the batch size decreases (90 training epochs with RandAugment, original learning rate schedule).

| Layer | Images / Worker | | | | | |
|---|---|---|---|---|---|---|
| | 128 | 64 | 32 | 16 | 8 | 4 |
| BN-ReLU | 77.6 | 77.6 | 77.1 | 76.7 | 74.8 | 72.0 |
| GN-ReLU | 76.9 | 77.1 | 77.0 | 77.0 | 77.0 | 77.2 |
| FRN | 77.2 | 77.1 | 77.2 | 77.1 | 76.8 | 77.2 |
| EvoNorm-S0 | **78.0** | 77.8 | 77.9 | **78.1** | **78.1** | 77.8 |
| EvoNorm-S1 | 77.9 | **77.8** | **78.0** | 77.8 | 77.8 | **78.0** |
| EvoNorm-S2 | 77.5 | 77.7 | 77.7 | 77.6 | 77.6 | 77.6 |

Table 11: ResNet-50 results as the batch size decreases (180 training epochs with RandAugment, original learning rate schedule).

| Layer | Images / Worker | | | | | |
|---|---|---|---|---|---|---|
| | 128 | 64 | 32 | 16 | 8 | 4 |
| BN-ReLU | 77.7 | 77.6 | 77.2 | 76.3 | 74.5 | 70.6 |
| GN-ReLU | 77.1 | 77.2 | 77.3 | 77.3 | 77.3 | 77.3 |
| FRN | 77.2 | 77.4 | 77.4 | 77.2 | 77.4 | 77.4 |
| EvoNorm-S0 | **78.3** | **78.3** | **78.4** | **78.3** | **78.2** | **78.2** |
| EvoNorm-S1 | 78.2 | 78.2 | 78.3 | 78.1 | 78.1 | **78.2** |
| EvoNorm-S2 | 78.2 | 78.1 | 78.0 | 77.9 | 77.9 | 78.0 |

Table 12: ResNet-50 results as the batch size decreases (180 training epochs with RandAugment, cosine learning rate schedule).

| Layer | Images / Worker | | | | | |
|---|---|---|---|---|---|---|
| | 128 | 64 | 32 | 16 | 8 | 4 |
| BN-ReLU | 73.3 | 73.2 | 72.7 | 70.0 | 64.5 | 60.4 |
| GN-ReLU | 71.5 | 72.6 | 72.9 | 72.5 | 72.7 | 72.6 |
| FRN | 73.3 | 73.4 | 73.5 | 73.6 | 73.5 | 73.5 |
| EvoNorm-S0 | **74.0** | **74.0** | 73.9 | 73.9 | **73.9** | 73.8 |
| EvoNorm-S1 | 73.7 | **74.0** | 73.6 | 73.7 | 73.7 | **73.8** |
| EvoNorm-S2 | 73.8 | 73.4 | 73.7 | **73.9** | **73.9** | 73.8 |

Table 13: MobileNetV2 results as the batch size decreases.

Figure 9: Selected samples from BigGAN-deep + EvoNorm-B0.

## E.1 Trade-off on Lightweight Models

Batch normalization is efficient thanks to its simplicity and the fact that both the moving averages and affine parameters can be fused into adjacent convolutions during inference. EvoNorms, despite being more powerful, can come with more sophisticated expressions. While the overhead is negligible even for medium-sized models (Table 5), it can be nontrivial for lightweight, mobile-sized models.

We study this subject in detail using MobileNetV2 as an example, showing that EvoNorm-B0 in fact substantially outperforms BN-ReLU in terms of both accuracy-parameters trade-off and

Figure 10: ImageNet accuracy vs. params and accuracy vs. FLOPs for MobileNetV2 paired with different normalization-activation layers. Each layer is evaluated over three model variants with channel multipliers $0.9\times$, $1.0\times$ and $1.1\times$. We consider the inference-mode FLOPs for both BN-ReLU and EvoNorm-B0, allowing parameter fusion with adjacency convolutions whenever possible.

accuracy-FLOPs trade-off (Figure 10). This is because the cost overhead of EvoNorms can be largely compensated by their performance gains.

## E.2 Stability Criterion

Here we empirically investigate whether the stability criterion is complementary to the quality criterion. While this additional rejection criterion is efficient, it may incur false positives hence lead to slower search progress and/or eventually lower accuracies of the best evolved layers. Figure 11 shows that this is not the case. In fact, the stability criterion speeds up the overall evolution progress by up to $2\times$ in this particular controlled experiment. One possible explanation is that layers that achieve high accuracies are also likely numerically stable.

Figure 11: Impact of the stability criterion on the evolution progress.

## E.3 Scale-invariance of EvoNorms

It is intriguing to see that EvoNorm-B0 keeps the scale invariance property of traditional layers, which means rescaling the input $x$ would not affect its output. EvoNorm-S0 is asymptotically scale-invariant in the sense that it will reduce to either $\frac{x}{\sqrt{s^2_{w,h,c/g}(x)}}$ or a constant zero when the magnitude of $x$ becomes large, depending on the sign of $v_1$. The same pattern is also observed for most of the EvoNorm candidates presented in Appendix D. These observations are aligned with previous findings that scale invariance might play a useful role in optimization [66, 67].