[Reviews · NeurIPS 2020]

Review 1

Summary and Contributions: This paper discusses an intriguing idea of searching for combined normalization + activation layer using primitive ops, including first and second moments in particular. The resultant layers (EvoNorms) is comparable for BatchNorm + ReLU for large batch size and is better than GroupNorm + ReLU for small batch size. The method is also validated on other applications such as instance segmentation and GAN.

Strengths: The paper focus on a novel and interesting problem: Can an improved normalization + activation layer be found via AutoML using primitive operations? It has brought a systematic and positive answer to this question. The batch dependent layer found (EvoNorm-B0) is slightly better than existing BatchNorm + ReLU variant, while the batch independent layer (EvoNorm-S0) is shown to be much better than existing GroupNorm + ReLU method when the batch size is small. The paper has some interesting analysis/observation on the challenge of applying AutoML to this problem (Line 134-148). In particular, the observation the solution found for one architecture does not generalize easily across other architectures leads to multi-architecture evaluation (Section 4.3) that are shown to make the method generalize well across architectures. This has certainly made the proposed solution more relevant in practice. The analysis of the properties of the EvoNorms reveal interesting similarities between the identified solution to existing design. It is fairly curious to see that the EvoNorm-B0 is similar to BatchNorm + ReLU but with a second moment across spatial dimensions, and EvoNorm-S0 is similar to Swish with some enhancements. The method is validated not only on image classification (ImageNet), but also on large-scale classification, but also on instance segmentation and GAN with similar improvements. These are very important as in those application the norm and activation functions usually need to be customized to get best performance.

Weaknesses: - Although this paper set forth to find normalization + activation from scratch, the search space itself (in Table 2) is fairly skewed towards existing design (for example, the moments are first and second moments). The identified solution looks a lot like a combination of existing approaches (BN, IN etc. in normalization, and ReLU/sigmoid for activation). A stronger version should consider expanding this search space and see if a truly different norm + activate layer emerges. - This paper is solid first step to attempt searching for norm + activation, but perhaps it should also consider finding a different layer for a different stage in the network. For example, EvoNorm-S0 depends on a form a group norm for sample statistics, would layers with different number of channels/spatial dimensions require a different number of groups for example for best accuracy? - The paper shows layers found from ImageNet classification can generalize. This does not break from existing manual approaches where the best architecture (including the norm and activation design) are identified first from image classification task. What could truly utilize the potential of AutoML and to find customized architecture for target task, say instance segmentation. Update: The author response sufficiently address my concerns. I remain very positive and excited about this paper. I agree with the author response that the paper shouldn't be considered as "yet another AutoML paper". In fact, the proposed idea of searching for the normalization layer - a basic building block of modern neural networks - is only sparsely considered before at best to my knowledge. And it makes a lot of sense considering the appropriate way to do normalization is quite important for real-world applications. I also agree that the improvements are indeed pretty large. This to me shouldn't be seen as a concern for this paper.

Correctness: The experiments and the analysis are insightful and seems correct. As this work is mostly empirical, the correctness of this work is contingent upon the experiment section. The paper seems fairly solid by using exactly the same training setting for comparison It is unclear to me exactly the how the architecture is evaluated, is it on a hold-out set separate from the test set where the numbers are reported?

Clarity: The paper is mostly well-written and filled with details.

Relation to Prior Work: The literature survey is comprehensive and seems to put this work correctly in the context. It is perhaps best if the relation to [50] can be expanded, as that work seems to share a very similar motivation (although less practical).

Reproducibility: Yes

Additional Feedback:


Review 2

Summary and Contributions: This paper leveraged the creativity of evolution to automate the design of normalization layers and activation functions together as a unified building block, whereas in previous works they are designed separately. This idea itself is novel. Authors also proposed novel rejection protocols (based on their quality and stability criteria) to improve the efficiency of the search by filtering out least promising candidates in each iteration. Those criteria might be also useful for other AutoML methods. Using their approach, authors discovered powerful normalization-activation layers (which they termed EvoNorms), and intensively tested them and showed that they outperform several most commonly used normalization-activation layers. Those EvoNorms layer might also inspire future design of normalization-activation layers.

Strengths: This approach is well motivated: normalization layers and activation functions are previously designed separately and this means there’s potential for AutoML to design them together to improve performance. Leveraged the creativity of evolution. Instead of handcrafting scalar-to-scalar transformations in the previous activation function search, they search instead for general tensor-to-tensor transformations. Experiment section is very comprehensive with many architectures including ResNets, MobileNetV2, MnasNet, EfficientNets, and transfer well to Mask R-CNN, SpineNet and BigGAN-deep and compared EvoNorms with many most commonly used normalization-activation layers. Paper is very well-written and clear.

Weaknesses: Lack of ablation study for the two rejection protocols is my mean concern and is the principle component of my rating. While the experiments focused intensively on various architectures and normalization-activation layers, it is not clear how those two rejection protocols contribute to the final results. Although both of them are very well motivated by the two observations, the observations themselve are not sufficient to justify the two rejection protocol. Evolution is extremely creative and the more constraint we manually put on it, the more we limit its creativity. More specifically, the search space for complex problems are usually very deceptive, for example, a candidate might be numerically unstable based on the stability criterion, however this candidate may have potential to be evolved into a surprisingly powerful one later on, but based on the current protocol it might be rejected early on. In table 3, random search with rejection also achieved very good results and authors’ EvoNorm only outperformed it by a small margin, which also concerns me about the effectiveness of the search method itself. Given enough computation and enough search, it’s only natural to discover better solutions. So that’s why I think it’s important to have a more detailed ablation study. To achieve the “strong generalization” mentioned in paper, authors used multi-objective optimization to explicitly optimize its cross-architecture performance. However, the architectures are still res-net based architectures and from the experiments in this paper it is still somewhat unknown how well EvoNorms perform on architectures that are more different with res-net architectures, whereas we know traditional methods such as BN-Relu can still work well. On CIFAR experiments the EvoNorms outperforms traditional methods significantly however when generalizing to ImageNet and other tasks the improvements are only marginal. A Minor point: In the abstract, before writing the acronym consider list the full phrase, like Neural Architecture Search (NAS) Updates: The ablation study were added and addressed my concern on the stability criterion, thanks! And I'd encourage authors to include the results and some discussion about it in the paper. All my other concerns are also addressed, so I'm increasing my score from 6 to 8.

Correctness: The methods are correct and the claims are backed up by a comprehensive set of experiments. Using their approach, authors discovered powerful normalization-activation layers (which they termed EvoNorms), and intensively tested them and showed that they outperform several most commonly used normalization-activation layers on a wide range of common architectures, including ResNets, MobileNetV2, MnasNet, EfficientNets, and transfer well to Mask R-CNN, SpineNet and BigGAN-deep, even though the improvement compared to traditional methods is only marginal.

Clarity: The paper is very well-written and clear.

Relation to Prior Work: For this paper, the relation to prior work is very clear: normalization layers and activation functions are previously designed separately and in this work they treat them as a unified building block and use evolution to search for the entire block. This work is different from prior work on neural architecture search in a sense that in previous neural architecture search, the search was about architecture based on well-defined building blocks while in this work they aim to discover new building blocks based on primitive mathematical operations. Although the motivation is similar to AutoML-Zero but authors’ method can actually achieve competitive results on large scale CV tasks.

Reproducibility: Yes

Additional Feedback: It’d be interesting to see more advanced evolutionary algorithms applied in this setting, for example, the novelty search based method originally by Kenneth Stanley, to overcome the deceptions in search.


Review 3

Summary and Contributions: The paper proposes a new normalization layer(EvoNorm) designed by neural architecture search. Authors use an evolutionary algorithm to automatically search a new normalization layer. To reduce enormous search costs, they select samples after short training, and propose two rejection protocols: 1. Reject samples with lower validation accuracy (quality evaluation), 2. Reject samples with higher gradient norm (stability evaluation). For strong generalization, samples are evaluated over three architectures. To reduce bias towards helping Resnet, they use Pareto frontier criterion. With extensive experimental results, authors showed strong generalization of the proposed normalization layer to other tasks (segmentation, GAN).

Strengths: The paper is well written. Although evolutionary algorithm requires a lot of time and computation cost, the authors showed various experimental results. This type of work to promote the strong generalization of search algorithm is meaningful in NAS research area (not just aiming to improve accuracy).

Weaknesses: To evaluate instability of training, using gradient norm is noteworthy, but the overall search algorithm lacks some novelty. it needs to explain their intuition or motivation to set 4 initial nodes (one input, one constant zero tensor, and two trainable vectors) in the DAG, because original batch normalization has no trainable vectors except beta and gamma.

Correctness: The paper is well written.

Clarity: Yes

Relation to Prior Work: Yes

Reproducibility: Yes

Additional Feedback: 1. It would be better to show details on search cost (GPU days, how many GPUs were used for search, etc). 2. Need more explanation on setting initial nodes in a DAG. 2. Batch normalization has no learning parameters (except gamma and beta), so I wonder why authors set two input nodes in DAG as trainable vectors (v_zero, v_one). It is not clear why authors set a constant zero tensor as one of initial nodes. -----Post rebuttal----------- Since the authors explained the novelty issues, the score is increased to 6. The significance of a NAS-based search of generalized normalization layer is now understood. The inclusion of 0, $v_0$, and $v_1$ in the computation graph is well explained.


Review 4

Summary and Contributions: This paper presents a methods of searching normalization-activation layers with 3 different designs selected for batch-dependent (as replacement for BatchNorm-ReLU) and sample-dependent (as replacement for GroupNorm-ReLU) respectively. It follows typical NAS pipelines. The searched norm-act layers, dubbed EvoNorms, show some interesting properties and comparable or slightly better performance than the counterparts in ImageNet classification, COCO detection and segmentation and BigGAN image synthesis.

Strengths: + It is an interesting idea of treating feature normalization and non-linear activation as a single tensor-to-tensor module. + The searched EvoNorms show some unconventional insights from the perspective of feature normalization and non-linear activation (eg, "some EvoNorms do not attempt to center the feature maps, and others require no explicit activation functions."). + The experimental results are promising.

Weaknesses: - The design of search space could be improved (Table 2). It introduces 12 element-wise ops and 3 aggregation ops (1st, 2nd moments). The searched EvoNorms heavily utilize the 3 aggregation ops. It will be more interesting to introduce more aggregation ops. - The search cost overhead is not cheap. It needs 2 days with 5000 CPU workers. It does not report how fast it could be if using GPUs. This may limit the applicability if the CIFAR-10 proxy task need to be changed. - The experimental results are not sufficiently thorough. The EvoNorms are searched jointly with three types of neural architectures (resnet-50, mobilenetv2, efficientnet-b0) with less number of channels used. First, the searched EvoNorms are not tested beyond the three architectures, e.g., resnext and densenets. So it is not clear if how applicable the EvoNorms are for neural architectures that are not included in the searching phase. Second, even for three architectures used in search, it will be better to see how the searched EvoNorms work for deeper variants such as resent-101. These two missing experiments are important since BatchNorm/GroupNorm-ReLU are generic and the performance gain of the reported experiments are not significantly high. - Some theoretical analyses will definitely help understanding the searched EvoNorms if presented. For example, some analyses could be done similar in spirit to "How Does Batch Normalization Help Optimization? by S. Santurkar et al".

Correctness: Yes, they seem to be correct based on the reviewer's understanding.

Clarity: The paper is easy to follow.

Relation to Prior Work: Yes, discussions on related work are sufficient.

Reproducibility: Yes

Additional Feedback: Post-rebuttal update: Thank the authors for their efforts in the rebuttal. Most of my concerns have been addressed. After going through other reviewers' comments, I think the paper has some interesting observations, but the cost over improvement may not be justified well. I slightly increase my overall score.

[Author Response · NeurIPS 2020]

**R2, R4: The improvements are marginal/slightly better.** Our gains are indeed large. Here're some highlights:

- EvoNorm-S0 is the state-of-the-art in the small batch size regime (Table 4), outperforming BN-ReLU by 7.8%
on ResNet-50 and by 7.3% on MobileNetV2.

- We achieve clear gains over other influential works such as GroupNorm (GN). On ResNet-50, EvoNorm-S0
outperforms GN-ReLU by 1.2% (large batch) and 1.1% (small batch) (Table 4).

- On Mask R-CNN, GN-ReLU improves BN-ReLU by 0.6mAP whereas EvoNorm-B0 improves BN-ReLU by
1.9mAP (Table 5). 1.9mAP gain on detection/instance segmentation is considered to be very significant in the
vision community (e.g., it is comparable with the gains of RetinaNet [1] over its previous models). [1]

We'd also like to emphasize that EvoNorms beat BN-ReLU on 12 (out of 14) different classification models/training
settings (Table 3 & 4) and all instance segmentation models (Table 5). B0 also improves over BN-ReLU on BigGAN in
terms of FID (Table 6). These are significant considering the predominance of BN-ReLU in ML models.

**R3: "the overall search algorithm lacks some novelty."** We argue that our work should not simply be evaluated as
"yet another AutoML paper" (with the expectation that some fancy search algorithms must be proposed), but rather under
the scope of designing new normalization-activation techniques which are central to deep learning (BatchNorm paper is
cited 20000 times, LayerNorm is cited 2000 times and GroupNorm is cited 700 times). While most existing norm-act
layers are in the form of $\frac{x - \text{mean}(x)}{\text{std}(x)}$ followed by a scalar-to-scalar transformation, we design highly unconventional
layers that perform equally well/better on a variety of important vision tasks (see EvoNorm-B0, which does not center
the feature maps or require explicit activation functions). The novelty of these discovered layers, as well as the new
insights, should not be diminished because they were discovered automatically instead of manually.

**R2, R4: Can EvoNorms generalize to deeper variants (e.g., ResNet-101) and architecture families not included
during search (e.g., DenseNet)?** The answer is yes. In Secs 5.1-5.3 we conducted extensive experiments to verify
that EvoNorms can perform well on previously unseen architectures (as well as tasks beyond classification), including
MnasNet, EfficientNet-B5, Mask R-CNN + FPN/SpineNet and BigGAN–none of them was used during search. Below
we also provide additional results on ResNet-101 and DenseNets (as requested by R4):

| Model | DenseNet-BC-121 | DenseNet-BC-169 | ResNet-101 | ResNet-101 + Mask R-CNN | Batch Independent? |
|---|---|---|---|---|---|
| BN-ReLU | 74.9 | 76.0 | 79.3 | 43.8 | No |
| EvoNorm-B0 | **75.8** | **76.4** | **79.5** | **45.1** | No |
| GN-ReLU | 74.5 | 75.8 | 79.0 | 44.0 | Yes |
| EvoNorm-S0 | **75.7** | **76.5** | **79.4** | **44.5** | Yes |

Table 1: ImageNet Accuracies & COCO box mAPs. ImageNet models were trained using the +aug+2×+cos setup with batch size 8×128. Like all the other experiments in the paper, we do not tune hyperparameters wrt EvoNorms.

**R3: Why including $0$, $v_0$, $v_1$ in the computation graph?** They are basic elements in popular activation functions.
E.g., nodes 0 and $v_1$ are required to form graphlets like $\max(x, 0)$ (ReLU) and $x\sigma(v_1 x)$ (SiLU/Swish), respectively.
Moreover, they can also be used by evolution in creative ways–see the expression of EvoNorm-B0.

**R1, R4: The search space design could be improved.** Our current search space contains 12 element-wise ops and
12 aggregation ops (3 types of statistical moments, each with 4 possible index sets). While expanding the search space
further is an interesting future direction, we believe the current search space is sufficient to demonstrate the concept
for the following reasons: (i) it is already challenging enough (Figure 2: none of 5000 random layers can outperform
BN-ReLU), and (ii) it is also interesting enough to yield highly novel design patterns, such as EvoNorm-B0.

**R1: Is there a hold-out set for layer search?** Yes. The layers were evolved on CIFAR-10 and then reranked using
10% of ImageNet's training set (L204-205 & L547-549 in Appendix D). The test set is never used during search.

**R2: Stability criterion might reject promising layers early on.** This is a valid
concern. If so, one would expect the stability criterion to eventually hurt the
accuracies of the best layers during evolution and/or slow down the search progress.
We provide an ablation study (as suggested by R2) in the figure on the right hand
side, showing that this is not the case. In fact, the stability criterion speeds up the
overall search progress by up to 2×. One possible explanation is that layers that can
achieve high accuracies are also likely numerically stable.

## Footnotes

[1] T.-Y. Lin, P. Goyal, R. Girshick, K. He, and P. Dollár. Focal loss for dense object detection. In CVPR 2017.


[Meta-Review · NeurIPS 2020]

The paper focuses on designing new neural architectures ; it presents a new search space and new optimization criteria. The new search space includes tensor-to-tensor operators integrating activation and normalization functions ; the criteria involve an early performance indicator (this is classical) and a stability indicator (this is new). The rebuttal addressed nearly all reviewers' concern: * about the significance of the performance gains; * about the generality of the approach when applied to other architectures; * about the fair evaluation (with a hold-out); * about the impact of the stability indicator (lesion study). Congratulations for the work ! The AC would like the computational cost of the evolution to be spelled out in the revised paper (beyond "a relatively large number of CPUs" ..); how many tournaments ? As a suggestion, it might be interesting to see whether (and how) scale insensitivity (E.2) could be used as a 3rd rejection criterion.